# A Nutritionally Complete Oral Nutritional Supplement Powder Improved Nutritional Outcomes in Free-Living Adults at Risk of Malnutrition: A Randomized Controlled Trial

**DOI:** 10.3390/ijerph191811354

**Published:** 2022-09-09

**Authors:** Suey S. Y. Yeung, Jenny S. W. Lee, Timothy Kwok

**Affiliations:** 1Department of Medicine and Therapeutics, Faculty of Medicine, The Chinese University of Hong Kong, Hong Kong, China; 2Department of Medicine, Alice Ho Miu Ling Nethersole Hospital, Hong Kong, China; 3Department of Medicine and Geriatrics, To Po Hospital, Hong Kong, China

**Keywords:** aging, malnutrition, oral nutritional supplement, community, quality of life

## Abstract

Background: This randomized controlled trial investigated the effectiveness of an oral nutritional supplement (ONS) on nutrition-related outcomes over 12 weeks in Chinese adults with or at risk of malnutrition. Methods: 88 Chinese adults ≥18 years living independently in Hong Kong with Mini Nutritional Assessment-Short Form (MNA-SF) score ≤11 were randomly assigned to (1) 2 servings/day of nutritionally complete ONS powder made with water (Fresubin^®^ Powder (Fresubin Kabi Deutschland GmbH, Bad Homburg, Germany), 600 kcal, 22.4 g protein) for 12 weeks (intervention group) or (2) no treatment (control group). The primary outcome was increase in body weight (BW) over 12 weeks. Secondary outcomes included improvement in body mass index (BMI), mid-arm circumference (MAC), calf circumference, MNA-SF score, quality of life, self-rated health, frailty, and diet quality. Results: The intervention group showed a significantly higher mean increase in BW compared with the control group (1.381 kg, intervention vs control, *p* < 0.001). The intervention group also showed significantly higher mean increases in BMI, MAC, calf circumference, intake of energy, protein, vitamin D, and calcium compared with the control group. No group differences in the changes of other outcomes were observed. Conclusions: For Chinese free-living adults at risk of malnutrition, daily consumption of a nutritionally complete ONS powder improved nutritional outcomes compared with the control group.

## 1. Introduction

Healthy aging, defined as “the process of developing and maintaining the functional ability that enables wellbeing in older age”, has been identified as a public health priority by the World Health Organization [1]. Today, the goal of public health is to improve physical function and psychological well-being and prevent geriatric syndromes such as sarcopenia [1,2,3]. A life course approach to healthy aging has also been suggested, indicating that lifestyle interventions such as nutrition should be provided in a timely manner in order to optimize health and well-being.

Inadequate nutrition contributes to weight loss and the development and progression of geriatric syndromes including osteoporosis, falls, frailty, and sarcopenia [4,5]. Across different health care settings, up to 33% are identified as malnourished, and up to 53% have a risk of malnutrition [6]. Malnutrition imposes a substantial economic burden on health care systems, with an estimate of USD 157 billion in the United States [7] and USD 66 billion in China [8]. This highlights the importance of identifying and managing malnutrition, particularly in the community, to reduce the risk of undertreating nutritional problems, which may continue into other settings such as hospitals and long-term care. Literature has shown the effectiveness of oral nutrition supplements (ONS) for improving nutritional status such as body weight, energy intake, and mid-arm circumference among individuals with or at risk of malnutrition [9,10,11]. However, previous trials mostly focused on individuals in hospitals and long-term care settings [12], or free-living individuals with specific chronic diseases or conditions [10]. In contrast, evidence in free-living individuals with or at risk of malnutrition is scarce, and Asian data on this research topic are limited [13,14].

The aim of this randomized controlled trial was to investigate the effectiveness of a nutritionally complete ONS powder for improving nutrition-related outcomes over 12 weeks in Chinese free-living adults with or at risk of malnutrition compared with a control group without any treatment.

## 2. Materials and Methods

### 2.1. Participants

Chinese men and women aged 18 years or older living independently in the community in Hong Kong were identified through local newspaper advertisements and through doctor and allied health referrals (through clinics at Prince of Wales Hospital, Shatin Hospital, Tai Po Hospital, and the Jockey Club Centre for Osteoporosis Care and Control) and word of mouth. Potential participants were invited for a brief screening using the Mini Nutritional Assessment-Short Form (MNA-SF) [15]. Those with a score of 11 or below on the MNA-SF were classified as at risk of malnutrition or malnourished and were invited for further screening to confirm eligibility. MNA-SF has been determined to be an accurate nutritional screening tool for detecting the risk of malnutrition across different settings [16]. Study exclusion criteria included recent (i.e., past 3 months) or concurrent participation in any clinical trial or dietary intervention program, self-reported allergy or intolerance to the ingredients of the nutrition supplement, tube feeding or parenteral nutrition, new use of medications started within three months (amphetamine and its derivatives, levothyroxine, biguanides), a diagnosis of cancer that is currently undergoing treatment, poorly controlled or unstable chronic diseases, existing gastrointestinal diseases or severely impaired gastrointestinal function, dysphagia or high aspiration risk, liver failure or severe renal insufficiency, neurological or psychiatric disorders, an unhealed bone fracture within the past 12 months, planned surgery or hospitalization during the study period, or major medical or psychological illness. The study protocol was approved by the Clinical Research Ethics Committee of the Chinese University of Hong Kong (CREC 2020.223) and registered at the US Clinical Trials Registry (NCT04474886). Written informed consent was obtained from all participants.

### 2.2. Study Design

This multi-center, open-labeled, randomized controlled trial took place between January 2021 and February 2022 in Hong Kong. A study coordinator generated a randomization list using a computer program (Statistical Package for the Social Sciences) before the commencement of the trial. After eligible participants completed the baseline assessment at the study site, they were randomly assigned into one of the two groups: nutrition supplement (intervention group) or no treatment (control group). Outcome measures were collected again at 12 weeks, whereas interim body weight measurements were also collected at 4 and 8 weeks.

### 2.3. Data Collection

After screening and eligibility to participate in the study was confirmed, the standardized questionnaire and measurements were completed at the scheduled visits. Data on demographic and lifestyle factors including age, sex, education level, marital status, employment status, length of residence in Hong Kong, co-existing diseases, medication, smoking habits, and alcohol use were collected at baseline. The primary outcome was increase in body weight over 12 weeks. Secondary outcomes included improvement in body mass index (BMI), mid-arm circumference (MAC), calf circumference, MNA-SF score, health-related quality of life, self-rated health, frailty status, and diet quality in terms of intake of energy and selected nutrients over 12 weeks.

Body weight and height were measured to the nearest 0.1 kg and 0.1 cm, respectively, using standardized methods. Two measurements were taken, and the average body weight and height were used to calculate BMI (body weight in kilograms divided by height in squared meters). BMI has been shown to have a U-shaped relationship with all-cause and cause-specific mortality in middle-aged and older Asians [17,18,19]. For Asians, the recommended normal BMI range is between 18.5 kg/m^2^ and 22.9 kg/m^2^ [20]. MAC was measured for the non-dominant arm, and calf circumference was measured for the left leg. Both circumferences were measured twice using a standardized protocol. Low muscle mass is one of the diagnostic criteria for sarcopenia [21]. Studies have shown that MAC and calf circumference were positively associated with muscle mass in older adults [22,23,24]. Given that the tools for measuring muscle mass such as dual-energy X-ray absorptiometry (DXA) and bioelectrical analysis (BIA) are costly and have limited accessibility, it has been suggested that simple and noninvasive anthropometric indices such as MAC and calf circumference can serve as practical proxy measures for muscle mass to diagnose sarcopenia in clinical settings [22,23,24]. The Asian Working Group on Sarcopenia in Older People 2019 has also recommended the use of calf circumference for identifying cases of sarcopenia [21]. MAC and calf circumference can reflect general functional ability, nutritional status, and current general health in older adults [25,26]. Moreover, MAC and calf circumference have been suggested as stronger predictors of mortality compared with BMI in older adults [26,27,28,29].

Health-related quality of life was assessed with the validated Chinese version of the 12-item Short Form Health Survey (SF-12), which includes physical and mental domains [30]. A higher score indicates better health-related quality of life in both domains. Self-rated health was assessed using a single question “Generally speaking, how is your health: excellent, very good, good, fair, or poor?” A higher score indicates worse self-rated health in this study. Frailty status was assessed using the 5-item FRAIL scale [31]. A score of 1 was assigned to each of the 5 components: fatigue, resistance (i.e., inability to climb up 10 steps), ambulation (i.e., inability to walk 2 to 3 blocks), having more than 5 diseases, and recent weight loss of more than 5%. A score of 0 on the FRAIL scale represents robust, 1 to 2 is pre-frail, and 3 to 5 is frail.

To assess their diet quality, participants were asked to complete a 3-day food record including food and beverage consumption on 2 weekdays and 1 weekend day. Instructions on how to estimate the portion sizes in household measures were given. At the scheduled visits for the baseline and 12-week assessments, the food records were checked for completeness and clarified by trained research personnel. Additional information about unclear items or portions was obtained, and examples of household measures were used to improve the estimation. Daily intakes of energy, macronutrients, and selected micronutrients were calculated using the Food Processor Nutrition analysis and Fitness software version 8.0 (ESHA Research, Salem, MA, USA), with the addition of the composition of local foods based on the food composition table from China and Hong Kong [32].

### 2.4. Sample Size Calculations

The sample size was calculated based on the primary outcome: increase in body weight over 12 weeks. A sample size of 31 participants per group was required to detect a mean difference of 1.7 kg in body weight between the intervention group and the control group using independent two-sample mean test at 80% power and 5% significance level based on a previous study with similar study design [33]. Considering a noncompliance rate of 15% for the consumption of nutritional supplements as well as a drop-out rate of 15% based on previous studies with ONS intervention among Chinese adults in nursing homes [34] and in the community in Hong Kong [35], a final sample size of 88 (44 × 2 groups) participants was required.

### 2.5. Intervention and Control Group

The study product is a nutritionally complete ONS powder made with water that provides either complete or supplementary nutritional support for adults with or at risk of malnutrition. Participants were asked to consume 2 servings (4.5 scoops/69 g powder per serving) of the study product (Fresubin^®^ Powder, also known as Fresubin^®^ Powder Fibre, (Fresenius Kabi Deutschland GmbH, Bad Homburg, Germany)) per day for 12 weeks. Two servings of the study product provide an additional 600 kcal, 22.4 g protein, 8 µg vitamin D, 400 mg calcium and multivitamins per day. The full nutrient composition of the study product is shown in Appendix A. Instructions on how to prepare the supplement were provided to the participants. They were also encouraged to consume the study products between meals. Due to the different needs or preferences of the participants, we offered participants flexibility and the option to prepare the ONS powder in two different caloric densities: (i) 2 times per day: each time 69 g powder + 140 mL water = 200 mL ready to use product (1.5 kcal/mL) or (ii) 3 times per day: each time 46 g powder + 160 mL water = 200 mL ready to use product (1 kcal/mL). Both options contributed 2 servings of the study product (i.e., a total of 138 g study product). Compliance rate of the study product, calculated by the servings of study product consumed as a proportion of that advised, was assessed by counting the number of empty and (partially) used cans returned every 4 weeks. The minimum acceptable compliance rate was set at 80% and above.

For the control group, no treatment or placebo was given. Participants were asked to maintain their usual physical activities and dietary habits during the 12-week study period and received the same measurements as the intervention group.

### 2.6. Adverse Events Associated with Nutritional Intervention

A telephone enquiry hotline was available during office hours for participants to report any adverse events. Regular telephone calls at 2, 6, and 10 weeks were made to all participants to record any adverse events.

### 2.7. Statistical Analysis

The analysis was conducted following the intention-to-treat (ITT) principle. Continuous variables are presented as means and standard deviations. Categorical variables are presented as numbers and percentages. Independent-sample *t*-tests or chi-square tests were used to compare values between groups at baseline and 12 weeks. Paired-sample *t*-tests were used to compare values within groups. For outcome measures collected at more than 2 time points (i.e., body weight and body mass index), differences between groups from baseline to 12 weeks were analyzed using linear mixed models, assuming random missing values driven by the variables used in the analysis. Time, treatment, and their interaction were set as fixed factors, and the participant was set as the random factor. For outcome measures collected at only 2 time points, differences between groups from baseline to 12 weeks were analyzed using independent-sample *t*-tests. Group differences in change from baseline after 12 weeks are presented as estimated mean difference and 95% confidence interval (CI). All statistical analyses were performed using the statistical package SPSS version 26.0 (IBM SPSS Statistics for Windows, IBM Corp. (Armonk, NY, USA)), and *p*-values of <0.05 were considered statistically significant (2-tailed).

## 3. Results

### 3.1. Characteristics of Participants

Figure 1 shows the study flow chart. Out of 424 individuals screened for inclusion, 123 were eligible, of whom 35 declined to take part, resulting in 88 being included in the trial and randomized to the intervention or control groups. All of the participants were classified as at risk of malnutrition according to the MNA-SF. Table 1 shows the baseline characteristics of enrolled participants. The mean age was 66.5 ± 9.7 years (age range between 41 to 89 years), and 87.5% were female. Most of the participants had an education level of secondary school or below (73.9%), were married (56.8%), and were not employed (81.8%). The mean length of residence in Hong Kong was 58 ± 14 years. The most common self-reported medical condition was bone, joint, or muscular problems (45.5%), followed by gastrointestinal problems (31.8%). Polypharmacy was present in 8% of the participants. None of the participants was a current smoker, and 3.4% of participants were current drinkers. The baseline characteristics of the two groups were comparable. A total of 6 participants (6.8%) dropped out of the study during the 12 weeks (2 in the intervention: medical decision (*n* = 1), dislike the taste of the study product (*n* = 1); 4 in the control group: medical decision (*n* = 2), COVID-pandemic (*n* = 1), unwilling to follow study procedure (*n* = 1)). Compliance with the study product was high in the intervention group, with 88% of the participants achieving at least 80% compliance. Two participants (4.5%) reported that the taste of the study product was too sweet, with one participant dropping out because of the taste while the other participant eventually adapted to the taste as her compliance rate increased from 87.5% at week 4 to 100% at week 12. Most of the participants consumed the study product between lunch and dinner (65.8%) and at breakfast (51.2%). More broadly, 39% of the participants consumed the study product between meals, while 61% consumed the study product with at least one main meal. Most participants (78%) preferred the study product in a concentrated caloric density (1.5 kcal/mL).

### 3.2. Study Outcomes

The baseline mean body weight and BMI of the participants were 43.6 ± 5.8 kg and 17.8 ± 1.6 kg/m^2^, respectively. Table 2 shows the outcome measures at baseline, the changes from baseline, and the estimated mean group differences in changes from baseline between the intervention and control groups. There were no significant differences between the two groups at baseline except for the MNA-SF score, which was slightly higher in the intervention group (10.7 ± 0.6 points) compared with the control group (10.3 ± 1.0 points) (*p* = 0.029). For the primary outcome, mean increase in body weight in the intervention group (1.78 ± 1.34 kg) was significantly higher compared with the mean increase in the control group (0.30 ± 0.87 kg); the corresponding estimated mean group difference was 1.381 kg, 95% CI 0.980 to 1.783, *p* < 0.001. Mean increases from baseline to 12 weeks in BMI (0.73 ± 0.54 kg/m^2^), MAC (0.75 ± 0.74 cm) and calf circumference (0.36 ± 0.54 cm) in the intervention group were significantly higher compared with those in the control group (BMI 0.11 ± 0.37 kg/m^2^, MAC −0.05 ± 0.43 cm, calf circumference 0.05 ± 0.37 cm); the corresponding estimated mean differences were 0.578 kg/m^2^ (95% CI 0.411 to 0.745, *p* < 0.001), 0.795 cm (95% CI 0.529 to 1.061, *p* < 0.001) and 0.313 cm (95% CI 0.111 to 0.516, *p* = 0.003) respectively. There was no significant group difference in the change of MNA-SF, physical domain and mental domain of SF-12, self-rated health and FRAIL scale (all *p* > 0.05).

Table 3 shows the energy and selected nutrient intakes at baseline, the changes from baseline, and the estimated mean group differences in changes from baseline between the intervention and control groups. At baseline, there were no significant differences between the groups in intake of energy, macronutrients, % energy from macronutrients, fiber, vitamin D, and calcium. At 12 weeks, the intervention group showed a significantly greater mean increase in daily intake of energy (309 ± 482 kcal), protein (11.9 ± 27.2 g), fat (14.5 ± 24.6 g), carbohydrate (29.6 ± 63.3 g), vitamin D (6.5 ± 3.3 µg), and calcium (207 ± 233 mg) compared with the mean changes in the control group (energy −33 ± 367 kcal, protein −0.8 ± 23.5 g, fat −1.9 ± 20.8 g, carbohydrate −4.7 ± 46.7 g, vitamin D 0.6 ± 3.6 µg and calcium −48 ± 275 mg); the corresponding estimated mean group differences were 342 kcal (95% CI 154 to 529, *p* = 0.001), 12.7 g (95% CI 1.5 to 23.8, *p* = 0.027), 16.4 g (95% CI 6.3 to 26.4, *p* = 0.002), 34.2 g (95% CI 9.8 to 58.6, *p* = 0.007), 5.9 µg (95% CI 4.4 to 7.4, *p* < 0.001) and 254 mg (95% CI 142 to 366, *p* < 0.001) respectively. There were no significant group differences in increasing fiber intake and % energy from macronutrients (all *p* > 0.05).

Appendix A shows the intakes of other micronutrients at baseline and at 12 weeks and the estimated mean group differences in changes from baseline between the intervention group and the control group. At baseline, no significant differences between the groups in intake of micronutrients were observed. At 12 weeks, the intervention group showed significantly greater increases in daily intakes of vitamin A, thiamin, riboflavin, niacin, vitamin B6, vitamin B12, biotin, vitamin C, vitamin E, pantothenic acid, chloride, copper, iodine, iron, magnesium, manganese, molybdenum, potassium, selenium, and zinc (all *p* < 0.005).

### 3.3. Adverse Events

No adverse events related to the consumption of the study product were recorded. A total of 6 participants withdrew from the study (*n* = 5 were not related to the study intervention).

## 4. Discussion

The present trial shows that the consumption of a nutritionally complete ONS powder for 12 weeks significantly improved body weight; BMI; MAC; calf circumference; and intakes of energy (from carbohydrates and fat), protein, and multiple micronutrients such as vitamin D and calcium among Chinese free-living adults at risk of malnutrition, compared with a similar group of subjects receiving no treatment. No significant effect on MNA-SF, health-related quality of life, self-rated health, or frailty was found.

The effectiveness of ONS consumption on weight gain is consistent with the literature [10,36]. A Cochrane review suggested the beneficial effects of ONS on body weight, with a mean 2.2% weight gain in malnourished individuals [36]. This is equivalent to an average weight gain of approximately 1 kg for an individual weighing 43 kg. The mean body weight gains were 1.78 kg in the intervention group and 0.30 kg in the control group. The observed weight gain was also slightly more than in a 12-week trial among individuals with or at risk of malnutrition living in a long-term care setting using a different study product of similar energy and protein contents (+1.2 kg in the intervention group and −0.5 kg in the control group) [33]. In our study, body composition was not measured to distinguish the composition of the weight gain. However, the weight gain observed in this study may reflect an increase in muscle mass as we found that the use of the study product for 12 weeks significantly improved MAC and calf circumference, which have been suggested as surrogate markers of muscle mass for diagnosing sarcopenia [21,22,23,24]. Muscle mass declines from the age of 30, and the rate of decline is even higher after the age of 60 [37]. Sarcopenia, characterized by low muscle mass, low muscle strength, and/or low physical performance, increases with age and significantly increases the risk of falls and fractures [38,39]. Nutritional strategy is one of the key interventions for preventing and treating sarcopenia [40]. The increases in MAC and calf circumference may reflect the potential benefits of the study product in muscle-related outcomes and its protective role against falls and fractures among individuals with or at risk of malnutrition. Of note, sex-specific risk factors for sarcopenia have been shown, with part of the literature suggesting that at risk of malnutrition predicted the progression to sarcopenia in females but not in males [41,42]. Given that the majority of our participants were females, our findings are promising and strengthen nutrition as a strategy for preserving muscle health in females with suboptimal nutritional status.

Although the present study did not find a significant group difference in the change in MNA-SF score, the consumption of the study product resulted in significantly greater intakes of total energy (carbohydrates, fat), protein, and multiple micronutrients such as vitamin D and calcium compared with the control group. Due to a lower intake of protein and reduced anabolic response to protein intake, the need for dietary protein is higher with increasing age [43]. Inadequate protein intake may lead to the loss of lean body mass, particularly muscle mass [44]. Furthermore, vitamin D is essential for muscle mass as it suppresses the expression of myostatin (an inhibitor of muscle growth) in muscle tissue [45]. Evidence has shown that protein and vitamin D have positive impacts on muscle mass and play a role in the prevention and management of sarcopenia [46,47,48,49]. Another common nutritional issue in middle-aged and older adults is low calcium intake [50,51,52]. Traditional dietary habits in Asian countries have been suggested as one of the barriers to increasing calcium intake, which may have implications for bone health [50]. As vitamin D is essential for calcium absorption [53], the additional calcium and vitamin D intake from the study product may help to increase the level of calcium intake and potentially improve bone health, particularly in the Asian population. Apart from its classic role in musculoskeletal health, vitamin D helps to regulate immune response, and better vitamin D repletion may contribute to reducing inflammatory problems [54]. The increase in dietary intake in the intervention group is of utmost importance, as it indicates that the study product may be effective in increasing the intake of key nutrients for middle-aged and older adults and potentially improving their musculoskeletal health and immune function, particularly when other nutritional care such as dietary counseling and food fortification are not sufficient to reach nutritional goals [55].

Subjective measures including health-related quality of life and self-rated health did not differ significantly between groups. However, it should be noted that smaller declines in the subjective measures were observed in the intervention group. Indeed, there was a trend towards statistical significance in the estimated mean group difference in the change in the score on the physical domain of the SF-12 (*p* = 0.063). The reason for the lack of significant differences in health-related quality of life and self-rated health between groups could be a relatively high quality of life and self-rated health reported before the study. Compared with a previous trial in Chinese community-dwelling individuals with sarcopenia, who had a mean of 40.3 in the physical domain of SF-12 and a mean of 49.0 in the mental domain of SF-12 [35], our participants had higher scores on both domains of the SF-12. Similarly, compared with a previous trial in apparently healthy community-dwelling Chinese middle-aged and older adults, who had a mean of 2.9 for self-rated health [56], our participants had higher scores for self-rated health. This may have resulted in little room for improving quality of life and self-rated health in our participants. Another potential reason is the impact of the COVID-19 pandemic. As our trial was conducted during the COVID-19 pandemic in Hong Kong, our participants may have experienced a decline in physical and mental health during this period. Furthermore, our nutritional intervention lasted for 12 weeks, which may have been too short for improving these outcomes. The lack of significant differences in subjective measures was similar to a study that reported no improvement in quality of life after 12 weeks of ONS among malnourished free-living individuals in the UK [13].

In our study, we did not find a significant improvement in frailty status after the intervention. This is somewhat in line with our expectations as most of our participants (63.2%) were robust at baseline. Furthermore, it has been suggested that multidomain interventions (physical exercise, nutritional, pharmacological, psychological and/or social interventions) tended to be more effective than mono-domain interventions on frailty status [57,58]. This could explain the lack of significant group differences in the change of frailty status in our participants.

### 4.1. Strengths and Limitations

The present study has several strengths. Our sample size provided sufficient power to detect a significant difference in the primary outcome. It is one of the few randomized controlled trials conducted in free-living individuals with or at risk of malnutrition. Data on dietary intake were available and assessed using a standardized method. The participants showed good compliance with the intervention, and no adverse effects with the use of the study product were reported. Several limitations should be acknowledged. First, body composition data were not available. However, MAC and calf circumference, which can be easily measured by most clinicians as it requires minimal undressing and costly equipment, have been proposed as reliable proxy measures for muscle mass. Second, the trial was conducted during the time of the COVID-19 pandemic in Hong Kong, which may have had impacts on the physical and mental health of the participants. Third, this study was neither placebo-controlled nor blinded. Fourth, plasma vitamin D level was not measured, and therefore, whether there was an improvement in plasma vitamin D status in the intervention group could not be determined. Furthermore, our participants were volunteers, and the majority of them were female, who were more likely to be health-conscious and motivated to comply with the study procedures; this may have added a bias to interpretations of the findings. Of note, although we aimed to evaluate the effectiveness of the ONS on nutrition-related outcomes in Chinese free-living adults aged ≥18 years, the recruited participants were generally above middle age, and 53.4% were ≥65 years old. The effectiveness of ONS on nutrition-related outcomes in the younger Chinese population may be even greater given the age-related differences in chronic diseases and muscle protein synthesis [43,59].

### 4.2. Implications for Clinical Practice

Our findings indicated that consumption of 2 servings of a nutritionally complete ONS powder daily for 12 weeks was able to improve nutritional outcomes in terms of body weight, BMI, MAC, calf circumference, and dietary intake. Among Chinese free-living adults with or at risk of malnutrition, consumption of the study product was popular between lunch and dinner and at breakfast. This consumption pattern of ONS may be recommended, given that a local study observed low protein intakes at breakfast among Chinese sarcopenic older adults and suggested the need to increase protein intake at breakfast [60]. The overall compliance with the study product was high, and middle-aged and older adults of Asian origin have well accepted milk-based nutritional supplements such as Fresubin^®^ Powder (Fresenius Kabi Deutschland GmbH, Bad Homburg, Germany). Given the potential consequences of malnutrition and associated economic burdens, nutritional risk should be identified and intervened early in the community. Clinicians may recommend the use of the study product among free-living individuals with or at risk of malnutrition in order to improve their nutritional outcomes and support healthy aging in terms of musculoskeletal health and immune function.

## 5. Conclusions

Overall, the results of this study support the potential benefits of a nutritionally complete ONS powder (Fresubin^®^ Powder, Fresenius Kabi Deutschland GmbH, Bad Homburg, Germany) for improving body weight; BMI; MAC; calf circumference, and intakes of energy, protein, and multiple micronutrients such as vitamin D and calcium among Chinese free-living adults at risk of malnutrition, which may potentially lead to improved muscle-related outcomes and support healthy aging. No significant group differences in the changes in MNA-SF, health-related quality of life, self-rated health, or frailty were observed. Further trials aimed at verifying the effectiveness of this nutritional supplement in mid- and long-term clinical outcomes are warranted.

## Figures and Tables

**Figure 1 ijerph-19-11354-f001:**
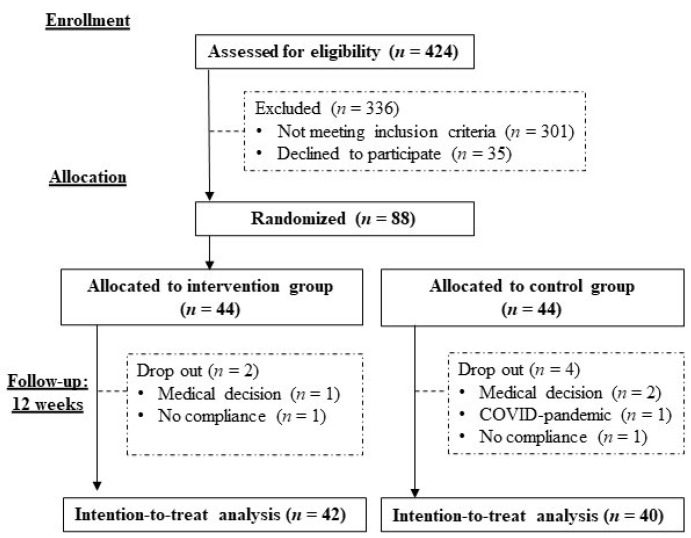
Study flow chart.

**Table 1 ijerph-19-11354-t001:** Baseline characteristics of enrolled participants.

	Intervention (*n* = 44)	Control (*n* = 44)	*p*-Value ^1^
Age, years	66.3 ± 9.5	66.7 ± 10.0	0.827
Sex, *n* (%)			0.747
Male	5 (11.4)	6 (13.6)	
Female	39 (88.6)	38 (86.4)	
Education level, *n* (%)			0.808
Secondary or below	33 (75.0)	32 (72.7)	
Tertiary or above	11 (25.0)	12 (27.3)	
Marital status, *n* (%)			0.085
Married	21 (47.7)	29 (65.9)	
Others ^2^	23 (52.3)	15 (34.1)	
Employment status, *n* (%)			0.269
Working	6 (13.6)	10 (22.7)	
Nonworking ^3^	38 (86.4)	34 (77.3)	
Length of residence in Hong Kong, years	58.2 ± 15.6	58.6 ± 13.4	0.895
Self-reported major medical history, *n* (%)			
Endocrinologic diseases	6 (13.6)	4 (9.1)	0.739
Cardiovascular diseases	13 (29.5)	6 (13.6)	0.070
Bone, joint, or muscular problems	22 (50.0)	18 (40.9)	0.392
Gastrointestinal problems/diseases	14 (31.8)	14 (31.8)	
Cancer	2 (4.5)	5 (11.4)	0.434
Polypharmacy, *n* (%) ^4^	3 (6.8)	4 (9.1)	1.000
Current smoker, *n* (%)	0	0	-
Current drinker, *n* (%)	1 (2.3)	2 (4.5)	1.000
Body mass index, kg/m ^2^	17.6 ± 1.6	18.0 ± 1.7	0.225
MNA-SF score, points	10.7 ± 0.6	10.3 ± 1.0	0.029
At risk of malnutrition, *n* (%)	44 (100)	44 (100)	-
Energy intake, kcal/d	1695 ± 497	1686 ± 355	0.929
Protein intake, g/d	77 ± 27	79 ± 26	0.684

Age, length of residence in Hong Kong, body mass index, MNA-SF score, energy and protein intake are presented as mean +/− SD, standard deviation. ^1^
*p*-value by independent *t* test or chi-square test or Fisher exact test where appropriate. ^2^ Others included single, widowed, separated, or divorced. ^3^ Nonworking included being a housewife, retired, or unemployed. ^4^ Polypharmacy represents having ≥5 medications. MNA-SF, Mini Nutritional Assessment-Short Form.

**Table 2 ijerph-19-11354-t002:** Outcome measures at baseline, changes from baseline, and the estimated mean group differences in changes from baseline between the intervention group (*n* = 42) and the control group (*n* = 40).

	Group	Mean ± SD	Change from Baseline, Mean ± SD	Between-Group Difference in Change from Baseline	
		Baseline	12 Weeks	Estimated Mean (95% CI) ^a^	*p*-Value
Body weight, kg	Intervention	43.2 ± 6.3	1.8 ± 1.3	**1.381 (0.980, 1.783)**	**<0.001**
	Control	44.0 ± 5.2	0.3 ± 0.9		
Body mass index, kg/m^2^	Intervention	17.6 ± 1.6	0.7 ± 0.5	**0.578 (0.411, 0.745)**	**<0.001**
	Control	18.0 ± 1.7	0.1 ± 0.4		
MAC, cm	Intervention	21.8 ± 2.1	0.8 ± 0.7	**0.795 (0.529, 1.061)**	**<0.001**
	Control	22.4 ± 1.9	0.0 ± 0.4		
Calf circumference, cm	Intervention	32.0 ± 2.0	0.4 ± 0.5	**0.313 (0.111, 0.516)**	**0.003**
	Control	31.5 ± 2.1	0.0 ± 0.4		
MNA-SF score, points	Intervention	10.7 ± 0.6	0.5 ± 0.7	−0.225 (−0.649, 0.199)	0.294
	Control	10.3 ± 1.0	0.7 ± 1.2		
SF-12—physical, score	Intervention	46.3 ± 10.0	1.6 ± 6.4	2.618 (−0.150, 5.385)	0.063
	Control	46.3 ± 7.6	−1.0 ± 6.2		
SF-12—mental, score	Intervention	53.7 ± 7.3	−0.7 ± 8.6	0.868 (−2.728, 4.464)	0.632
	Control	52.9 ± 7.3	−1.6 ± 7.7		
Self-rated health, score	Intervention	3.5 ± 0.8	0.0 ± 0.9	−0.073 (−0.402, 0.257)	0.662
	Control	3.6 ± 0.9	0.0 ± 0.6		
FRAIL scale, score	Intervention	0.5 ± 0.9	−0.3 ± 0.6	−0.062 (−0.359, 0.235)	0.679
	Control	0.6 ± 0.9	−0.2 ± 0.7		

CI, confidence interval; MAC, mid-arm circumference; MNA-SF, Mini Nutritional Assessment-Short Form; SF-12, 12-item Short Form Health Survey. Data are presented as mean ± SD. Bold values represented *p* < 0.05. ^a^ Represents group difference (intervention group minus control group) in 12-week change estimate (12-week minus baseline). Estimated mean (95% CI) was derived from linear mixed model for body weight and body mass index and from independent-sample *t*-test for the other outcomes.

**Table 3 ijerph-19-11354-t003:** Energy and selected nutrient intakes at baseline, changes from baseline, and the estimated mean group differences in changes from baseline between the intervention group (*n* = 42) and the control group (*n* = 40).

	Group	Mean ± SD	Change from Baseline, Mean ± SD	Between Group Difference in Change from Baseline	
		Baseline	12 weeks	Estimated Mean (95% CI) ^a^	*p*-Value
Energy, kcal/d	Intervention	1695 ± 497	309 ± 482	**342 (154, 529)**	**0.001**
	Control	1686 ± 355	−33 ± 367		
Protein, g/d	Intervention	77 ± 27	12 ± 27	**12.7 (1.5, 23.8)**	**0.027**
	Control	79 ± 26	−1 ± 24		
Fat, g/d	Intervention	63 ± 25	14 ± 25	**16.4 (6.3, 26.4)**	**0.002**
	Control	67 ± 19	−2 ± 21		
Carbohydrate, g/d	Intervention	207 ± 65	30 ± 63	**34.2 (9.8, 58.6)**	**0.007**
	Control	194 ± 45	−5 ± 47		
% kcal from protein	Intervention	18.2 ± 3.3	−0.5 ± 3.5	−0.8 (−2.5, 0.9)	0.337
	Control	18.7 ± 3.8	0.3 ± 4.0		
% kcal from fat	Intervention	33.5 ± 6.4	1.1 ± 6.4	1.3 (−1.5, 4.1)	0.352
	Control	35.7 ± 6.5	−0.2 ± 6.1		
% kcal from carbohydrate	Intervention	49.2 ± 7.9	−1.4 ± 8.7	−0.9 (−4.4, 2.6)	0.608
	Control	46.3 ± 6.6	−0.5 ± 7.2		
Fiber, g/d	Intervention	17 ± 9	3 ± 10	3.4 (−0.0, 6.8)	0.051
	Control	16 ± 7	−1 ± 5		
Vitamin D, µg/d	Intervention	2.7 ± 2.2	6.5 ± 3.3	**5.9 (4.4, 7.4)**	**<0.001**
	Control	2.6 ± 2.0	0.6 ± 3.6		
Calcium, mg/d	Intervention	604 ± 280	207 ± 233	**254 (142, 366)**	**<0.001**
	Control	595 ± 259	−48 ± 275		

CI, confidence interval. Data are presented as mean ± SD. Bold values represented *p* < 0.05. ^a^ Represents group difference (intervention group minus control group) in 12-week change estimate (12-week minus baseline). Estimated mean (95% CI) was derived from independent-sample *t*-test.

## Data Availability

The data presented in this study are available on request from the corresponding author. The data are not publicly available due to privacy and by agreement with the funder.

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
