# Peer review of "A Nutritionally Complete Oral Nutritional Supplement Powder Improved Nutritional Outcomes in Free-Living Adults at Risk of Malnutrition: A Randomized Controlled Trial"

_ijerph, 2022, doi:10.3390/ijerph191811354_

Round 1

Reviewer 1 Report

The paper responds to the needs of conducting scientific research of nutritional habits and general welfare among the free-living adults at risk of malnutrition. The paper is focused on investigating and describing the linkage between nutritional habits, oral nutritional supplement and other various aspects of living conditions, both mental and material.

Remarks

1. The study group consists mainly of women, which should be emphasized in the discussion of the results.
2. The research scheme is missing (flow chart).

Reviewer 2 Report

In their manuscript “A nutritionally complete oral nutritional supplement powder improved nutritional outcomes in free-living adults at risk of malnutrition: A randomized controlled trial”, the authors analyze the efficacy of a specific oral nutritional supplement in preventing/treating the malnutrition and in the achievement of some secondary objectives.

The manuscript is interesting, but I have to suggest some changes and explanations.

I would suggest specifying in the title and in the manuscript the name of the commercial product that the authors used for the study, given that the results concern that product only and not all ONPs.

I would suggest to cancel the term “PREDEFINED” outcomes in the abstract, and to be explicit about your secondary outcomes.

Controls: usual care. I would suggest not to use the term care or to explain what was the care (in what the care consisted?)

I would also suggest to use the term INCREASE instead of CHANGE of BMI because in your study all subjects were undernourished or at risk of malnutrition.

I don’t think you should specify that both weight and BMI increased because the increase of body weight automatically reflects on BMI given that your study was accomplished in 12 weeks and not after many years when the height might have changed.

You should distinguish from the very beginning between subjects at risk and those already undernourished. You should inform about their mean BMI at baseline and at the end and on how the final results differed between the two.

More details about nutritional assessment and diets prescribed are needed: did you assess them after the initial screening and what tools/measures you used? Did you give the same amounts of supplements to subjects who were only at risk of malnutrition and those who had severe malnutrition and why? How did you assess their intake? By food diaries? Explain it in the Methodology.

You should also specify in Methodology what was the minimum acceptable compliance rate.

I was surprised to find terms “fraility and older” and then read that you enrolled subjects older than 18. How come you did not decide to enroll only subjects older of 65 or at least adults? I think that you should consider this observation also because of your comments in discussion (line 340). I don’t find surprising that result if the age of subjects enrolled is considered.

Line 146: I would suggest removing of the observation “It can be prepared to two different caloric densities” because it is misleading. It may induce to think that subjects enrolled were assuming different densities. And I would suggest to explain why all subjects were assuming the same densities, independently from their nutritional status (at risk and undernourished).

Please do add something about cut off values of BMI for the Asiatic population. BMI of 18 is severe malnutrition in Europe, so it is difficult to understand for me what does the improvement from 17 kg/m2 to 18 kg/m2 means in your population.

I think that it would be better to change the order of tables. I would suggest to give the information about the nutritional status of subjects at baseline (BMI, n. subjects at risk, n. undernourished, protein calorie intake and needs), then you should explain in what the increase consisted and how did it reflect on BMI and other variables after 12 weeks.

If the compliance was 80%, it is obvious that subjects from the intervention group increased their intake. Why did the control group reduce it?

You should comment the protein intake in the older although I would suggest not to use a lot the words older and frailty because the subjects enrolled in your study are border (only 65 years mean age).

Line 297: I don’t think that your results are comparable with results of that study.

Round 2

Reviewer 2 Report

I think that you did a great job and that your manuscript may be pubblished now.